

# Molecular fingerprinting of particulate organic matter as a new tool for its source apportionment: changes along a headwater drainage in coarse, medium and fine particles as a function of rainfalls

Laurent Jeanneau[1*]; Richard Rowland[2]; Shreeram Inamdar[2]

[1] OSUR, UMR 6118 Géosciences Rennes, Université de Rennes 1 – CNRS, Rennes, France

[2] Water Science & Policy Graduate Program, University of Delaware, Newark, USA

*Correspondence to*: Laurent Jeanneau (laurent.jeanneau@univ-rennes1.fr)

**Abstract.** Tracking the sources of particulate organic matter (POM) exported from catchments is important to understand the transfer of energy from soils to oceans. The suitability of investigating the molecular composition of POM by thermally assisted hydrolysis and methylation using tetramethylammonium hydroxide directly coupled to gas chromatography and mass spectrometry is presented. The results of this molecular fingerprint approach were compared with previously published elemental (%C, %N) and isotopic data ($\delta^{13}$C, $\delta^{15}$N) acquired in a nested headwater catchment in Piedmont region, Eastern United States of America (12 and 79 ha). The concordance between these results highlights this molecular tool as a valuable method for source fingerprinting of POM. It emphasizes litter as the main source of exported POM at the upstream location (80 ± 14 %) with an increasing proportion of stream bed (SBed) sediments remobilization downstream (42 ± 29 %), specifically during events characterized by high rainfall amounts. At the upstream location, the source of POM seems to be controlled by the maximum and median hourly rainfall intensity. An added-value of this method is to directly investigate chemical biomarkers and to mine their distributions in term of biogeochemical functioning of an ecosystem. In this catchment, the distribution of plant-derived biomarkers characterizing lignin, cutin, and suberin inputs were similar in SBed and litter, while the proportion of microbial markers was 4 times higher in SBed than in litter. These results indicate that SBed OM was largely from plant litter that has been processed by the aquatic microbial community.



## 1 Introduction

Particulate organic matter (POM) plays key-roles in aquatic ecosystems, controlling the transfer and the bioavailability of energy, nutrients and micropollutants. The flux of POM from soils to oceans has been estimated at 0.2 GtC per year (Ludwig et al., 1996) with 80 % coming from biospheric inputs and the complement from petrogenic inputs (Galy et al., 2015). Assuming that the energy provided by natural organic matter is equivalent of the energy provided by the combustion of wood, this flux of POM corresponds to an energy of 2.8 EJ, that is to say less than 2 days of the global energy consumption of 2015 (yearbook.enerdata.net). This export mainly occurs during storm events, those hot moments being responsible for up to 80% of annual particulate organic carbon (POC) export depending on the investigated catchment (Dhillon and Inamdar, 2013; Jeong et al., 2012; Jung et al., 2012; Oeurng et al., 2011).

Among these hot moments, extreme events, defined as storm flow exceeded less than 10 % of the time (IPCC, 2001), seem to play a dominant role. In two contrasted catchments, a mountainous one in South-Korea and a lowland one in the Eastern United States of America (USA), the specific POC flux (flux per unit area of the catchment) has been shown to be non linearly related to total rainfall with a threshold value beyond which the slope increased sharply (Dhillon and Inamdar, 2013; Jung et al., 2014). The threshold value (approx. 70 mm in the American catchment and approx. 120 mm in the South-Korean catchment) and the magnitude of this increase differed between both catchments and are probably watershed-dependant. Is the non linearity of the relationship between rainfall amount and POC export observed previously linked to a modification of the source of POM? POM in a river system is a combination of allochthonous and autochthonous OM. The former is derived mainly from the soils and banks erosion, while the latter can be composed of fresh aquatic living organisms and bed sediments. The balance between these different sources is controlled (i) by the catchment' size and morphology and (ii) by the rainfall event characteristics (Tank et al., 2010).

Tracking the sources of POM can be done indirectly by investigating the sources of suspended matter. This can be done through the analysis of fallout radionuclides such as Beryllium-7, Lead-210 and Cesium-137 (Ritchie et al., 1974; Wallbrink and Murray, 1996; Walling, 1998) or by geochemical fingerprinting of rare elements (Collins and Walling, 2002). It can also be done directly by investigating the composition of POM using bulk-scale descriptors such as OC and Nitrogen concentrations, C/N ratio and stable isotopes $\delta^{13}C$ and $\delta^{15}N$ (Fox and Papanicolaou, 2008). Molecular biomarkers analyses have also been used. They are based on specific molecular classes such as lipid or lignin biomarkers (Goñi et al., 2013; Jung et al., 2015). Thermochemiolysis using tetramethylammonium hydroxide coupled to gas chromatography and mass spectrometry has already been applied to the investigation of the fate of river DOM (Jeanneau et al., 2015) and POM (Mannino and Harvey, 2000). This analytical technique is widely used to investigate the biogeochemistry of soil organic matter (Derenne and Quénéa, 2015) and, coupled to a principal component analysis (PCA), it has been shown to be valuable for forensic soils applications (Lee et al., 2012). An advantage of such an analysis is to generate a distribution of more than hundred identified target compounds with small amount of particulate matter (from 5 to 10 mg) (Jeanneau et al., 2014), giving a dataset rich enough to differentiate between sources (Walling, 2013). Here this analytical approach is combined



with a principal component analysis (PCA) to determine the main sources of POM as a function of the sediment size, the catchment size and the rainfall characteristics.

The first objective of this paper is to test the suitability of molecular biomarkers derived from THM-GC-MS as a tool to determine the sources of river POM. The second objective is to investigate how the sources of POM changed as a function of the catchment size, particle size of the sediment, and the hydrological characteristics of the rainfall events. This study is based on a subset of samples used to investigate the sources of POM exported during storm events using $^{13}$C and $^{15}$N as tracers (Rowland et al., 2017). We hypothesized that molecular biomarkers provide important insights into sources of POM and can be used as complimentary tracers for POM alongside or in addition to stable isotopes.

## 2 Material and methods

### 2.1 Site description

This study was conducted in a 79 ha watershed (second order stream) located in the Piedmont physiographic region of Maryland, USA (Figure 1). The watershed drains into the Big Elk Creek which discharges into the Chesapeake Bay. For a detailed description of the study site, refer to Rowland et al. (2017). Briefly, the watershed is predominantly forested with pasture along the outer periphery. Dominant canopy species include *Fagus grandifolia* (American beech), *Liriodendron tulipifera* (yellow poplar), and *Acer rubrum* (red maple). Bedrock formations consist of metamorphic gneiss and schist and soils are coarse loamy, mixed, mesic lithic inceptisols on slopes and oxyaquic inceptisols in saturated valley bottoms. Elevations in the watershed range from 77 to 108 m with slope gradients ranging from 0.16 to 24.5° (mean 6.3°). Mean annual precipitation from 1981 to 2010 in this region was 1173.5 mm, with late spring and late summer as the wettest and driest periods, respectively, and mean annual temperature is 13°C (Delaware State Climatologist Office Data Page, 2016).

### 2.2 Watershed monitoring and sampling strategy

Detailed information on monitoring and sampling is provided in Rowland et al. (2017). Climatological data was obtained from a local station maintained by the Delaware Environmental Observing System approximately 450 m from the 79 ha catchment outlet. This consists of temperature and GEONOR gage hourly rainfall measurements. Stream discharge estimates were obtained at 20-minute intervals using a Parshall flume at 12 ha stream location (nested within the 79 ha watershed, Figure 1) and a discharge rating curve calculated from paired pressure transducer and acoustic Doppler velocity meter measurements at a rectangular concrete culvert at the 79 ha location.

Suspended sediments were collected using in-situ samplers made of 10 cm diameter capped PVC pipes placed vertically in the middle of the stream. The upstream face of the pipes was perforated with 1.5 cm diameter holes beginning ~10 cm above the stream bed. During periods of elevated discharge, stream stage rose above the perforations, trapping suspended sediment within the sampler. The trapped sediment thus represented a time-integrated composite sediment sample (CSS). All CSS were retrieved within 24 hours of the end of an event and frozen prior to processing and analysis.

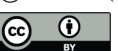



Seven potential sediment sources were identified within the catchment and have been sampled at three locations to integrate
their spatial heterogeneity (Rowland et al., 2017). These included the stream bed (SBed), exposed stream bank A (BaA) and
B (BaB) horizons, valley-bottom wetland surficial soils (W), forest floor litter (Li) and humus (FH) and the upland A
horizons (Up). Sampling was conducted during the summer of 2015. 500-750 g of each end-member were sampled using an
ethanol-cleaned trowel or auger from both of the main tributary branches of the watershed. Stream beds were sampled from
areas without major backwatering or pooling, as POM may undergo diagenesis here, and were composited along a three by
three-point grid within the channel. Bank sediments were collected from exposed incised banks with three points composited
from the A and B horizons. Forest floor litter and humus, valley-bottom wetland soils and upland A horizons samples were
composited from five points along 20 m transects in low gradient locations in order to integrate their spatial heterogeneity.
End-member soil and sediment samples and CSS were dried in acid-cleaned Pyrex dishes in an oven at 45°C until visibly
dry. Oven-dry CSS samples were partitioned into coarse (CPOM) > 1000 µm, medium (MPOM) 1000-250 µm and fine
(FPOM) < 250 µm size classes via dry sieving. Dry masses were recorded for particle size class from which the fractional
mass percent was calculated for each class in each CSS sample. End-member samples were pre-sieved at 2 mm to remove
large organic debris such as roots. Aliquots were lyophilized overnight and preserved in a desiccator cabinet until elemental,
isotopic and molecular analyses. CSS and end-member samples were pulverized and homogenized using a ceramic mortar
and pestle that was cleaned with ethanol between samples.

## 2.3   Analytical methodology

For elemental and isotopic analyses, please refer to Rowland et al., (2017). The thermochemiolysis using
tetramethylammonium hydroxide (TMAH) coupled to gas chromatography and mass spectrometry (THM-GC-MS) was
performed according to Jeanneau et al. (2014). Briefly we introduced approximately 5 mg of freeze-dried solid residue into
an 80 µL aluminum reactor with an excess of solid TMAH (ca. 10 mg) and 10 µl of a solution of dihydrocinnamic acid d9
(CDN Isotopes, ref. D5666) diluted at 25 µg/ml in methanol as an internal standard. The THM reaction was performed on-
line using a vertical micro-furnace pyrolyser PZ-2020D (Frontier Laboratories, Japan) operating at 400°C. The products of
this reaction were injected into a gas chromatograph (GC) GC-2010 (Shimadzu, Japan) equipped with a SLB 5MS capillary
column in the split mode (60 m × 0.25 mm ID, 0.25 µm film thickness). The temperature of the transfer line was 321°C and
the temperature of the injection port was 310°C. The oven was programmed to maintain an initial temperature of 50°C for 2
minutes, then rise to 150°C at 15°C min$^{-1}$, and then rise to 310°C at 3 °C min$^{-1}$ where it stayed for 14 minutes. Helium was
used as the carrier gas, with a flow rate of 1.0 ml/min. Compounds were detected using a QP2010+ mass spectrometer (MS)
(Shimadzu, Japan) operating in the full scan mode. The temperature of the transfer line was set at 280°C, the ionization
source at 200°C, and molecules were ionized by electron impact using an energy of 70 eV. The list of analyzed compounds
and m/z ratios used for their integration are given in the supplementary materials (Table S1). Compounds were identified on
the basis of their full-scan mass spectra by comparison with the NIST library and with published data (Nierop et al., 2005;



Nierop and Verstraten, 2004). They were quantified assuming similar ionization and detection efficiencies between all
compounds. This assumption means that the concentrations must be handled as rough estimations.
Target compounds were classified into four categories: low molecular weight organic acids (LOA), phenolic compounds
(PHE) including lignin and tannin markers, carbohydrates (CAR) and fatty acids (FA). The peak area of the selected m/z
(mass/charge) for each compound was integrated and corrected by a mass spectra factor (MSF) calculated as the reciprocal
of the proportion of the fragment used for the integration and the entire fragmentogram provided by the NIST library (Table
S1). The proportion of each compound class was calculated by dividing the sum of the areas of the compounds in this class
by the sum of the peak areas of all analyzed compounds expressed as a percentage. The analytical uncertainty for this
analytical method, expressed as a relative standard deviation ranged from 10 to 20% depending on the samples and the target
compounds. The use of THM-GC-MS to investigate the sources of POM meant that it was necessary to assume that matrix
effects are equivalent for all analyzed compounds in all samples.
**2.4   Statistical analyses and calculation of the proportions of the main sources of POM in CSS**
Statistical analyses were performed using XLSTAT (version 19.01, Addinsoft). First a principal component analysis (PCA)
was performed using the end-members as individuals and CSS as additional individuals. The relative proportions of the 112
target compounds and the sum of their concentrations in ng/mg of freeze-dried matrix were used as variables. The relative
distribution of target compounds allows the direct comparison of the different samples without concentration effect, while
using the sum of their concentrations takes into consideration the fact that the concentration of target compounds differed
from a sample to another.
The first PCA allows identifying the correlated variables on the basis of a modulus of the Pearson coefficient > 0.9. When
two variables were correlated, the least abundant was removed. Then a second PCA was performed. The variables with a
correlation lower than 0.4 with the two first factors (F1: 29.8%; F2: 17.2% of variance) were removed, resulting in a new set
of 71 variables. A third PCA was calculated and a hierarchical ascendant classification (HAC) was calculated using the
coordinates of the individuals (end-members and CSS) on the 9 first factors that explained 90.5% of the variance of the
dataset. This HAC identified Upland soils and Stream bank sediments as minor contributors. Consequently a fourth PCA
was calculated removing Upland soils and Stream bank sediments from the potential end-members. Similarly to the three
previous PCA, CSS were considered as additional individuals. The coordinates of CSS on the two first factors (on 10) of this
PCA (F1: 40.1%; F2: 24.0% of variance) were used to calculate the proportion of the three main sources of POM in CSS
identified as 1. stream bed sediments, 2. litter and 3. forest floor humus + wetland soil, resolving a system of equations with
three unknowns. To solve this system, the coordinates of end-members must be specified. The heterogeneity of the
distribution of target compounds resulted in an area for each end-member. To calculate the proportions and uncertainties, the
coordinates of end-members were randomly selected ten times in the areas defined by the 95% IC. When the calculation
gave a negative contribution for an end-member, it was set at 0 and the two others contributions were recalculated to sum at
100. Finally the contributions of those three sources were approximated for the bulk POM by using the proportion and the



OC content of each fraction. From the third PCA to the end of the procedure, this treatment was also performed adding TOC,
$\delta^{13}$C and $\delta^{15}$N from Rowland et al. (2017) as variables.
In order to test the efficiency of the source apportionment calculated with the molecular data, the proportions of end
members and their isotopic values (Rowland et al., 2017) were used to model the $\delta^{13}$C of CSS. Modeled values were
compared to measured values reported by Rowland et al. (2017) by calculating the relative standard deviation (RSD) and
against a linear regression model.

## 3    Results

### 3.1    Rainfall and hydrology

The molecular composition of POM in coarse, medium and fine size classes was investigated for four events. The rainfall
and discharge characteristics recorded for those events are indicated in Table 1. The total rainfall ranged from 40.1 (E4) to
148.9 (E1) mm, the maximum hourly rainfall (Imax) ranged from 19.9 (E1) to 31.3 (E3) mm h$^{-1}$ and the median hourly
rainfall (Imed) ranged from 0.4 (E3) to 2.2 (E2) mm h$^{-1}$. The maximum discharge for those events ranged from 15.6 (E4) to
150.1 (E1) l s$^{-1}$. Then the four events can be distinguished as follows. E1 was characterized by high rainfall, a low maximum
intensity (Imax), a mean median intensity (Imed) and a mean antecedent precipitation index (API7). E2 was characterized by
mean total rainfall, a mean Imax, a high Imed and a mean API. E3 was characterized by high rainfall and Imax, low Imed
and high API7. Finally E4 was characterized by low rainfall and Imax, a high Imed and a dry antecedent conditions (API7 =
0 mm). E2 and E4 were comparable in terms of precipitation regime but can be differentiated by the API7, E4 occurring
after 7 days without precipitation.

### 3.2    Size distribution

CSS were separated into coarse (>1 mm), medium (>250 μm) and fine (<250 μm) fractions, with the exception of CSS at the
downstream (79 ha) location for the fourth event (Table 1). In the 12 ha sub-catchment, the coarse, medium and fine
fractions represented 22 ± 20, 22 ± 4 and 55 ± 21 % of particulate matter, respectively, while in the 79 ha catchment, they
represented 61 ± 19, 22 ± 10 and 18 ± 10 % of particulate matter, respectively. In the 12 ha sub-catchment, the relative
standard deviation (RSD) of those proportions was 90, 17 and 37 % for the coarse, medium and fine fractions, respectively,
while in the 79 ha catchment it was 31, 45 and 55 %, respectively.

### 3.3    Molecular composition of end-members

The number of detected target compounds ranged from 49 (SBed#1) to 112 (FH). A Dixon test for extreme value identified
the lowest value (SBed#1) as an outlier (*p*-value = 0.011). Once this value removed, the number of detected target



compounds ranged from 75 (BaB) to 112 (FH). The low value recorded for one of the SBed could be due to a combination of
a low OC content with a low analytical efficiency. This sample was removed from the dataset.
The distribution of target compounds into chemical families gives a first overview of the molecular composition of OM in
the different end-members (Figure 2). In W, Li and FH, the main compounds are PHE and high molecular weight FA (> $C_{20}$,
HMW) that represent more than 30% of target compounds. In BaA and BaB, the proportion of PHE was lower ($22 \pm 4$ and
$19 \pm 1$ %, respectively; mean $\pm$ SD) than in W, Li and FH and the proportion of LMW FA was higher ($27 \pm 17$ and $35 \pm 9$ %,
respectively). In Up, compared to W, Li and FH, the proportion of HMW FA increased ($57 \pm 19$ %), while the proportion of
PHE decreased ($13 \pm 8$ %). In SBed, the main identified target compounds were LMW FA ($72 \pm 8$ %), while PHE and HMW
FA represented $15 \pm 2$ % and $9 \pm 4$ %, respectively.
HMW FA was composed of linear $n$-alkanoic acids from $n$-$C_{20:0}$ to $n$-$C_{32:0}$ with an even-over-odd predominance
characteristic of plant-derived inputs (Eglinton and Hamilton, 1967), linear ω-hydroxyacids and α,ω-diacids from $n$-$C_{16}$ to $n$-
$C_{28}$, 10,16-dihydroxy$C_{16:0}$ and 9,10,18-trihydroxy$C_{18:0}$ characteristic of plant-derived aliphatic biopolymers cutin and suberin
(Armas-Herrera et al., 2016; Kolattukudy, 2001). These two latter hydroxyacids were the main compounds among HMW
FA. The proportion of ω-hydroxyacids and α,ω-diacids among HMW FA is higher in roots than in leaves and can be used to
differentiate between suberin from roots and cutin from shoots (Mueller et al., 2012). This proportion decreased from soils
(Up, FH and W) and bank sediments to litter and was minimal for SBed ($17 \pm 8$ %), highlighting that the proportion of cutin
decreased from SBed, Li to bank sediments and soils.
PHE was composed of methoxy-benzene, -acetophenone, -benzaldehyde and -benzoic acids. These compounds derived from
lignin and tannins and are characteristic of plant-derived OM. The main compounds were guaiacyl-like structures: 3,4-
dimethoxybenzaldehyde, 3,4-dimethoxybenzoic acid methyl ester, *erythro* and *threo*-1,2-dimethoxy-4-(1,2,3-
trimethoxypropyl)benzene and syringil-like structures: 3,4,5-trimethoxybenzaldehyde and 3,4,5-trimethoxybenzoic acid
methyl ester, which is typical of the THM-GC-MS of OM deriving from woody plants (Challinor, 1995). Benzoic acid was
not classified in PHE since it was negatively (slope of the linear regression model: -0.20; -0.18; -0.17) and poorly correlated
(Pearson coefficient, $p$-value: 0.14, 0.002; 0.14, 0.002; 0.21, <0.001) with 3,4-dimethoxybenzoic acid methyl ester, 3,4,5-
trimethoxybenzoic acid methyl ester and 3-(3,4-dimethoxyphenyl)prop-2-enoic acid methyl ester, respectively, that are the
main representatives of the three types of lignin units analyzed by THM-GC-MS (Challinor, 1995). As a consequence, it was
not considered to calculate the proportion of molecules coming from lignins and tannins.
LMW FA included $n$-alkanoic acids from $n$-$C_{6:0}$ to $n$-$C_{19:0}$, *iso* and *anteiso* $C_{13:0}$, $C_{15:0}$ and $C_{17:0}$, *iso* $C_{14:0}$ and $C_{16:0}$ and $n$-
alkenoic acids $n$-$C_{16:1}$ and $n$-$C_{18:1}$. The LMW FA with less than 13 C atoms can derive from microbial or plant-derived
inputs, while the LMW FA with more than 13 C atoms are known as phospholipid fatty acids (PLFA) and are microbial
biomarkers (Frostegård et al., 1993). The proportion of microbial markers among target compounds was calculated
according to Jeanneau et al. (2014). It increased from litter and soils (<15%) to bank sediments ($18 \pm 12$ % and $25 \pm 7$ % in
BaA and BaB, respectively) to SBed ($48 \pm 15$ %).



### 3.4 Molecular composition of stream suspended sediments

The distribution of target compounds into the five chemical families previously described changed with the catchment size as illustrated on Figure 3. At the 12 ha location, this distribution was fairly homogenous across the particle classes. When averaged across size fractions and events, the THM-GC-MS of the POM of CSS sampled at the 12 ha location mainly produced PHE (48 ± 6 %, mean ± SD) and HMW FA (22 ± 10 %). The relative standard deviation weighted by the proportion (RSDp) was 13, 14 and 22 % for C, M and F fractions, respectively, which highlights a low inter-event variability of this distribution. At the 79 ha location, the distribution of target compounds was dominated by LMW FA (41 ± 20 %) and PHE (37 ± 9 %). It was almost stable between the three size fractions with a higher proportion of LMW FA in the M fraction. However, the RSDp was 50, 55 and 23 % for C, M and F fractions, respectively, which means a higher inter-event variability than at the 12 ha location.

### 3.5 End-members contributions

A hierarchical ascendant classifiction (HAC) was performed using the coordinates of end-members and stream sediments (CSS) on the nine first factors (90.5 % of variance) of the PCA, which were calculated with the relative proportions of target compounds and the sum of their concentrations as variables. Three classes were isolated. The first one included the three Li, one FH and one W as end-members, the size fractions of CSS from the 12 ha location and 3 size fractions of CSS from the 79 ha location. The second group included two W, two FH and the three BaA, BaB and U end-members. Finally the third group included the SBed end-members and the size fractions of CSS from the 79 ha location. Based on this HAC, U, BaA and BaB were considered as minor contributors to the POM exported from the 12 ha and 79 ha locations.

An additional PCA was then calculated using SBed, Li, FH and W as individuals, CSS as additional individuals, and the previously defined list of 71 variables. The two first factors of this PCA explained 64.1 % of the variance of this final dataset. The projection of end-members and CSS on the plan obtained with these two factors is illustrated on Figure 4. This projection allows differentiating: (i) the three groups of end-members, Li, SBed and a combination of FH and W, denoted FH-W and (ii) POM from the two sampling locations. Moreover the size classes were also separated. From this 2D projection, an area was defined for each end-member corresponding to the 95% CI. The results of the source apportionment calculated using this 2D projection are listed in Table 2.

At the 12ha location, as an average of the four sampled events, from FPOM to CPOM, the proportion of OM coming from SBed decreased from 17 ± 16 % (mean ± SD) to 1 ± 1 %, the proportion of OM coming from FH-W decreased from 16 ± 16 % to 8 ± 12 % and the proportion of OM coming from Li increased from 67 ± 7 % to 90 ± 11 %. The large uncertainties quantified by the mean RSD (78 ± 53 %, mean ± SD, n = 9) reflected the inter-storm variability of this source apportionment. Bulk POM was mainly inherited from Li with contributions ranging from 65 to 92 %.

At the 79ha location, as an average of the four sampled events, CPOM was mainly inherited from Li (63 ± 28 %) and SBed (36 ± 30 %). MPOM was mainly due to SBed inputs (49 ± 39 %) and received a substantial contribution of FH-W (17 ± 31



%). Similarly to CPOM, FPOM was mainly inherited from Li (55 ± 15 %) and SBed (38 ± 24 %). Similarly to the source apportionment at the 12ha location, the large uncertainties (RSD = 97 ± 57 %, n = 9) were due to inter-storm variability. Bulk POM was mainly inherited from Li with contributions ranging from 42 to 89 % and SBed with contributions ranging from 8 to 57 %.

## 4    Discussions

### 4.1    What are the main sources of POM for the watershed?

The HAC identified four main end-members for the stream water POM: litter (Li), the surface horizon of forest soils (FH) and wetland soils (W) and stream bed sediments (SBed). Li was the main source of POM identified along the catchment representing 80 ± 14 % and 49 ± 24 % of the POM exported from the 12 ha and 79 ha catchments, respectively. These high proportions of Li-derived POM is in accordance with the results of Jung et al. (2015) where isotopic and *n*-alkanes fingerprints of POM exported from a mountainous forested headwater catchment highlighted similarities with litter and surface soils. Moreover the decrease in the proportion of Li-derived OM along the catchment fits well with the observation of Koiter et al. (2013) where the contribution of topsoil sources of suspended sediments decreased from 75 to 30 % when moving downstream.

Stream bank A and B horizons and the surface horizons of upland soils did not group with any CSS, which would mean that they were minor contributors for the investigated samples. This seems to be in contradiction with the documented impact of bank erosion on the mobilization of particulate organic matter (Adams et al., 2015; Nosrati et al., 2011; Tamooh et al., 2012). This apparent contradiction could be due to the catchment's size. Contrary to the previously cited investigations (Adams et al., 2015; Nosrati et al., 2011; Tamooh et al., 2012), this present study focused on a headwater catchment (0.79 km²). In these small catchments, POM mainly comes from the erosion of surrounding soils as observed for monsoon floods in Laos (Gourdin et al., 2015; Huon et al., 2017) or from a combination of bedrock and surface erosion in an Alpine catchment with relative proportions controlled by the precipitations (Smith et al., 2013). However, in this catchment, the mobilization of stream banks has been shown to be effective in winter due to freeze-thaw process (Inamdar et al., under review). This present study analyzed four events sampled in spring and summer. The lower contribution of stream bank erosion could then be due to seasonal variability.

The relative proportion of LIG compared to HMW FA plotted against the proportion of α,ω-diacids and ω-hydroxyacids with more than 20 C atoms among HMW FA resulted in a visual differentiation of Li and SBed from W, FH, BaA and BaB and from Up (Figure 5). This observation highlights Li as the main origin of SBed plant-derived OM, which fits well with the high proportion of Li-derived POM in CSS from both catchments. Moreover from Li to SBed, (i) the ratio of coumaric and ferulic acids to vanillaldehyde, acetovanillone and vanillic acid, commonly noted C/V, decreased from 0.79 ± 0.26 to 0.20 ± 0.07, denoted that lignins were more biodegraded in SBed than in Li and (ii) the proportion of microbial markers





among the target compounds increased from 12 ± 5 to 48 ± 15 %. Both of these observations highlight the recycling of
terrestrial plant-derived OM in river sediments from a headwater catchment, and are in accordance with the higher
mineralization rate of soil organic carbon in river sediments (Wang et al., 2014).
**4.2 Are molecular data in accordance with isotopic and elemental data?**
A four-step analysis was performed to determine if the molecular data produced by THM-GC-MS were in accordance with
the isotopic results (Rowland et al., 2017) previously acquired on those samples.
The first one consists in a point-by-point comparison of the source apportionments resulting from the two approaches. Four
main observations were reported by Rowland et al. (2017) using the isotopic approach. First, "the litter layer was a dominant
contributor to CPOM, especially for the upstream locations". This is in agreement with our data: the proportion of Li-derived
CPOM was 90 ± 11 % and 63 ± 28 % for the 12ha and the 79ha catchments, respectively. Secondly, "the proportional
contributions of SBed and banks to MPOM and FPOM increased downstream". This is also in agreement with molecular
data, however stream banks were not considered as a main contributor through the present statistical treatment. The
proportion of SBed-derived POM increased from 8 ± 8 % to 49 ± 39 % and from 17 ± 16 % to 38 ± 24 % between the 12 ha
and the 79 ha catchments in MPOM and FPOM, respectively. Thirdly, "no appreciable shift was observed in CPOM source".
This is partly in agreement with the molecular data. The main contributor to CPOM was Li in the two locations but the
proportion of SBed-derived CPOM increased downstream. Finally, the highest contribution of forest floor humus was
observed in MPOM and FPOM for E4. This is in agreement with the source apportionment in this study since the proportion
of FH-W-derived POM was the highest for this event in CPOM, MPOM and FPOM from the 12 ha catchment and in MPOM
and FPOM from the 79 ha catchment.
In a second step, the quality of the source apportionment calculated from the end member mixing approach was investigated
by modeling the $\delta^{13}$C of the samples using the isotopic fingerprint of end members. These modeled values were compared to
the measured values used in the isotopic fingerprinting approach (Rowland et al., 2017). The relative standard deviation was
1.1 ± 0.2 % (mean ± 95% CI; n = 20) and the linear regression resulted in a slope of 1.01 (R² = 0.58; *p*-value < 0.0001;
Figure S1) highlighting a fairly good agreement between the model and the data, that is to say between the source
apportionment using molecular data and measured $\delta^{13}$C.
In a third step, TOC, $\delta^{13}$C, $\delta^{15}$N and C/N were added as variables in the PCA treatment. In a first PCA, W, FH, Li, SBed,
BaA, BaB and Up were considered as potential end members. A HCA using the nine first PCA factors (90.4 % of the
variance) highlighted BaA, BaB and Up as minor contributors, similarly to this step performed on molecular data alone.
Then a second PCA was calculated with FH, W, Li and SBed as potential end members and the CSS as additional
individuals. The two first factors represented 64.4 % of the variance and resulted in a clear differentiation between Li, SBed
and FH-W. The same approach was then applied using the molecular data alone, resulting in the calculation of the
proportions of those three end members in the CSS for ten different combinations of the position of end members in the 2D



plan created by the two first factors of the PCA. For each CSS sample a set of ten values was created for Li-, SBed- and FH-
W-derived POM (Table S2). Student T-test was used to compare these distributions between the modality "molecular data"
and the modality "molecular + isotopic, elemental data". A $p$-value was calculated for each sample. They ranged from 0.08
to 0.49 (0.25 ± 0.03; mean ± 95% CI), highlighting that there were no significant differences between the two approaches
(Table S3).
The final step aimed at investigating to what extent the molecular data are representative of bulk POM. The linear regression
between the sum of the concentrations of target compounds (expressed in µg/g of dry solid) and the total organic content
(expressed in % of dry solid) resulted in a correlation coefficient of 0.94 ($p$-value < 0.0001; Figure S2). This correlation
between bulk scale and molecular analyses has already been highlighted for sedimentary and dissolved OM (Jeanneau and
Faure, 2010; Jeanneau et al., 2014) and emphasizes the suitability of molecular investigations to determine the sources of
OM.
Once validated by this four-step comparison, what are the insights provided by the molecular approach on the source
apportionment of CPOM, MPOM and FPOM along this Piedmont headwater catchment?

## 4.3  Modification of the source apportionment as a function of rainfall parameters

These present results may be valuable to investigate the relationships between the sources of exported POM and rainfall
characteristics. However they have been acquired on only four events and this part of the discussion should be enriched by
future investigations.
Rainfall is the primary driver for C export since it controls soil erosion and stream discharge (Raymond and Oh, 2007).
Rainfall amount and API7 have been shown to control the export of POC from headwater catchments (Dhillon and Inamdar,
2013, 2014; Jung et al., 2014). Moreover Imax and Imed have also been identified as important drivers for soil erosion since
they control the rainfall erosivity (Wischmeier, 1959). The four investigated events represented a range of rainfall amounts,
Imax, Imed and API7.
Linear regression were performed between the proportions of Li-, SBed- and FH-W-derived POM in CPOM, MPOM and
FPOM from both catchments against rainfall amount, Imax, Imed and API7 (Table 1). With only four investigated events,
only relationships characterized by Pearson coefficient higher than 0.8 were considered. $p$-Values were not calculated for
those regressions since they would not have had any statistical value. With only four events the highlighted relationships
must be handled with care and may be seen as guidelines for future works.
In the 12 ha catchment, SBed-derived OM was positively related to Imax and API7 and negatively related to Imed. The
positive relationship with API7 was recorded in C and F fractions, while the positive relationship with Imax and the negative
relationship with Imed were recorded only in the F fraction. In the M fraction, SBed-derived OM was related to the total
rainfall. However since this fraction represented 22 ± 4 % (mean ± SD) of the exported particles, this relationship was not
considered as representative. In the 12 ha catchment the export of SBed-derived OM would be favored by rainfall
characterized by high Imax occurring after a period of dryness (Figure 6a). Moreover the proportion of FH-W-derived OM



was positively related to Imed in F fraction. This fraction represented $55 \pm 21$ % (mean $\pm$ SD) of the exported particles,
giving some representativity to this observation. A deeper analysis of the relationship between Imed and the proportion of
FH-W-derived OM in the different fractions from the 12 ha catchment highlights a concomitant control of API7 (Figure 6b).
For similar Imed (E2 versus E4), the proportion of FH-W-derived OM increased in the three fraction with dry antecedent
conditions. The activation of the soil reservoir seems to be controlled by both Imed and API7, which could be interpreted as
the necessity of a dry period to replenish a stock of soil OM available for soil erosion and that intensive and regular rainfalls
could result in higher soil erosion.
In the 79 ha catchment, the proportions of Li and FH-W were negatively related to the rainfall amount and the proportion of
SBed was positively related to this variable. These relationships were recorded in the C and M fractions, with the exception
of FH-W (only in the C fraction). A deeper analysis of the link between the POM source apportionment and the rainfall
amount highlights different threshold values for C, M and F fractions (Figure 6c). In M and F fractions, there was a sharp
modification of the source of POM between E4 (40.1 mm) and E2 (43.9 mm). The proportion of FH-W-derived POM
decreased from $64 \pm 20$ % to $0 \pm 1$ % and from $21 \pm 22$ % to $1 \pm 2$ %, in the M and F fractions, respectively. These decreases
were concomitant with increases in the proportion of SBed-derived POM from $0 \pm 0$ % to $43 \pm 8$ % and from $2 \pm 2$ % to $48 \pm$
$9$ %, in the M and F fractions, respectively. The source apportionment of FPOM remained unchanged by further increases of
the rainfall amount, while for MPOM the source apportionment was clearly modified during E1, which was characterized by
the highest rainfall amount (148.9 mm). The proportion of Li-derived POM decreased to $0 \pm 1$ % and the proportion of
SBed-derived POM increased from $58 \pm 9$ % to $95 \pm 7$ %. The source apportionment of CPOM drastically changed between
E2 and E3 (97.4 mm). The proportion of Li-derived POM decreased from $95 \pm 8$ % to $47 \pm 9$ % and the proportion of SBed-
derived POM increased from $2 \pm 4$ % to $53 \pm 9$ %. This source apportionment remained unchanged between E3 and E1.
Since the C fraction was the most important during events 1, 2 and 3, its source apportionment was an important driver of the
source of total POM. It was mainly modified between events 2 and 3 with a decrease in the proportion of Li-derived POM
and an increase in the proportion of SBed-derived POM. From these observations, the threshold value of 75 mm previously
found in this catchment with an increase in the slope of the POC exported in kg/ha as a function of the rainfall amount
(Dhillon and Inamdar, 2013) falls in the range from 43.9 mm (E2) to 97.4 mm (E3), where the main modifications of the
source of POM exported from the 79 ha catchment were observed. The increase in the proportion of SBed-derived POM
accompanied with the increase in the proportion of the C fraction could be the result of the exceeding of a threshold value of
the hydrodynamism for sediment remobilization.

### 4.4 Benefits and limitations of this molecular fingerprinting approach

The present molecular fingerprinting method has benefits and limitations. Among the benefits, when the analysis is
performed on-line, that is to say, when the products of the THM are directly sent to the GC, then the analysis needs low
sample mass, in the order of 5 to 10 mg. Then this method is based on the molecular composition of OM, which is perfectly
suitable to investigate the fate of POM. Moreover it takes advantage of the differences of chemical composition between



living organisms (microorganisms versus plants) and in their different parts (leaves versus roots). As a consequence the
recorded modifications can be discussed in term of biogeochemistry of POM.
However limitations must be considered. Seasonal variability of the molecular fingerprint could exist especially for quickly
reactive reservoir such as litter (Williams et al., 2016). In soils, the turnover of OM takes time (> 50 years; Frank et al.,
2012). Consequently their molecular fingerprints may be less sensitive to seasonal variations, with the exception of
agricultural soils subject to changes in vegetation cover. This limitation can be easily avoided by sampling the most reactive
end-members at different seasons. The second and third limitations come from the method itself. First this is a time-
consuming method because each compound must be determined with care in each sample. For an analysis, approximately
two hours are mandatory. Finally, because it is not only a value given by an analytical tool, using it asks having an expertise
in organic geochemistry.
When benefits and limitations are well considered, this molecular fingerprinting approach may be particularly suitable to
investigate the sources of POM in combination with other fingerprinting approaches.

## 5    Conclusion

This study emphasizes the suitability of molecular analysis of POM using THM-GC-MS to investigate the sources of POM
in headwater catchments. This analytical technique needs less than 5 mg of freeze-dried matter, which makes it realistic in
regard of the amount of suspended sediment exported and simple with only freeze-drying as a preparing step. With
approximately hundred of target compounds, the provided chemical fingerprint allows for the differentiation of the main
sources of exported POM, specifically between litter, surface soils, and in-channel sediments. The fairly good relationships
obtained by comparison with the conclusions gained by the isotopic-elemental investigation provide additional evidence in
favor of this organic fingerprinting approach. The present data highlight plant litter as the main source of exported POM with
an increasing contribution of stream bed sediments downstream. This latter contribution seems to be controlled by the
rainfall amount with a threshold phenomenon already observed for quantitative data. The contribution of soil erosion could
be controlled by both the median intensity of rainfall and the amount of rain in the previous 7 days. The investigation of
additional events in different catchments will be necessary to determine if those results are generic.

**Data availability**

Data are available on request from the corresponding author.

**Acknowledgements**

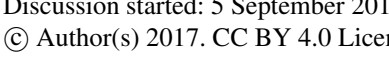
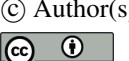


This study was funded by NSF ESPCoR Grant # IIA 1330238 (NEWRnet) and USDA NIFA Grant # 2015-67020-23585.
We would like to thank the Fair Hill Natural Resources Management Area for allowing us to conduct this study in the Fair
Hill Nature Preserve. Many thanks to students who assisted with sampling including Erin Johnson, Catherine Winters,
Chelsea Krieg, Shawn Del Percio, Margaret Orr and Daniel Warner.

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

**Figure captions**
Figure 1: Location of the study watershed in the Piedmont region of Maryland. Composite suspended sediments were
sampled at the 12 and 79 ha locations (grey circles). The sites of collection of end-members are indicated with triangles:
violet for Wet, blue for Bsed, green for FH and Li, orange for Up and yellow for BaA and BaB.
Figure 2: Relative proportions of low organic acids (LOA), phenolic compounds (PHE), low molecular weight and high
molecular weight fatty acids (LMW and HMW FA) and carbohydrates (CAR) among identified target compounds in the end
members. Uncertainties correspond to standard deviation of sampling triplicates (duplicates for SBed).
Figure 3: Relative proportions of low organic acids (LOA), phenolic compounds (PHE), low molecular weight and high
molecular weight fatty acids (LMW and HMW FA) and carbohydrates (CAR) among identified target compounds in the
coarse, medium and fine fractions of CSS. Uncertainties correspond to the inter-event standard deviation.
Figure 4: Plan defined by the two first factors of the PCA calculated using the distribution of target compounds. Squares
represent end members Li (green), FH-W (red) and Sbed (blue). The area charateristic of each end member is defined by the
95% confident interval. Circles represent CSS from the 12 ha (orange) and the 79 ha (purple) locations. The mean positions
for each size fraction are represented by large circles and uncertainties correspond to inter-event standard deviation.
Figure 5: 2D plot illustrating the variability of the distribution of plant-derived markers using the relative proportion of PHE
against HMW FA and the proportion of a,w diacids and wOH fatty acids among HMW FA (denoted HMW FA ratio).
Figure 6: Illustration of the most significant correlations between the source apportionments performed using the molecular
data and rainfall characteristics. At the 12 ha location, positive correlations (a) between the proportion of Sbed-derived POM
and Imax and (b) between the proportion of FH-W-derived POM and Imed. At the 79 ha location, positive correlation





between  Sbed-derived POM and rainfall amount (c). Coarse, medium and fine fractions are depicted by the dark grey, light
grey and white circles, respectively and the composite POM by the black diamond.





Table 1. Rainfall characteristics, discharge and proportion of coarse, medium and fine fractions for the 4 investigated storm events.

| | Event 1 | Event 2 | Event 3 | Event 4 |
|---|---|---|---|---|
| | *May 1, 2014* | *Apr. 21, 2015* | *July 3, 2015* | *Sept. 30, 2015* |
| ***Rainfall*** | | | | |
| total (mm) | 148.9 | 43.9 | 97.4 | 40.1 |
| max (mm h$^{-1}$) | 19.9 | 20 | 31.3 | 20.2 |
| median (mm h$^{-1}$) | 1.3 | 2.2 | 0.4 | 2.1 |
| API7 (mm) | 9.7 | 10.4 | 68.2 | 0 |
| | | | | |
| ***Discharge (12 ha catchment)*** | | | | |
| max (l s$^{-1}$) | 150.1 | 68.3 | 87.4 | 15.5 |

| ***Particle size distribution*** | | | | | | | | |
|---|---|---|---|---|---|---|---|---|
| | *12 ha* | *79 ha* | *12 ha* | *79 ha* | *12 ha* | *79 ha* | *12 ha* | *79 ha* |
| Coarse (%) | 52 | 81 | 20 | 43 | 12 | 58 | 6 | nd |
| Medium (%) | 22 | 13 | 22 | 32 | 27 | 20 | 18 | nd |
| Fine (%) | 27 | 7 | 59 | 25 | 61 | 21 | 75 | nd |





Table 2. Source apportionment calculated using the molecular data.

| | | 12 ha location | | | 79 ha location | | |
|---|---|---|---|---|---|---|---|
| | | Li (%) | Sbed (%) | FH-W (%) | Li (%) | Sbed (%) | FH-W (%) |
| **Event 1** | C | 97 ± 7 | 1 ± 2 | 3 ± 7 | 45 ± 9 | 55 ± 9 | 0 ± 0 |
| *May 1, 2014* | M | 78 ± 7 | 18 ± 5 | 4 ± 8 | 0 ± 1 | 95 ± 7 | 4 ± 7 |
| | F | 76 ± 12 | 13 ± 6 | 11 ± 16 | 48 ± 9 | 52 ± 9 | 0 ± 0 |
| | POM | 92 ± 9 | 4 ± 4 | 4 ± 11 | 42 ± 6 | 57 ± 8 | 0 ± 2 |
| **Event 2** | C | 95 ± 8 | 2 ± 3 | 3 ± 8 | 95 ± 8 | 2 ± 4 | 3 ± 8 |
| *Apr. 21, 2015* | M | 94 ± 9 | 2 ± 3 | 4 ± 9 | 57 ± 9 | 43 ± 8 | 0 ± 1 |
| | F | 69 ± 16 | 15 ± 6 | 17 ± 20 | 51 ± 10 | 48 ± 9 | 1 ± 2 |
| | POM | 86 ± 11 | 6 ± 4 | 8 ± 12 | 89 ± 9 | 8 ± 7 | 3 ± 4 |
| **Event 3** | C | 96 ± 5 | 3 ± 4 | 1 ± 5 | 47 ± 9 | 53 ± 9 | 0 ± 0 |
| *July 3, 2015* | M | 87 ± 6 | 10 ± 5 | 3 ± 7 | 42 ± 9 | 58 ± 9 | 0 ± 0 |
| | F | 61 ± 8 | 39 ± 7 | 0 ± 1 | 45 ± 11 | 51 ± 9 | 4 ± 6 |
| | POM | 81 ± 6 | 17 ± 6 | 2 ± 4 | 46 ± 10 | 53 ± 9 | 2 ± 2 |
| **Event 4** | C | 73 ± 22 | 0 ± 0 | 27 ± 22 | fraction not available | | |
| *Sept. 30, 2015* | M | 70 ± 22 | 0 ± 0 | 30 ± 22 | 36 ± 20 | 0 ± 0 | 64 ± 22 |
| | F | 62 ± 23 | 0 ± 0 | 38 ± 23 | 77 ± 20 | 2 ± 2 | 21 ± 22 |
| | POM | 65 ± 23 | 0 ± 0 | 35 ± 23 | - | - | - |





**Figure 01**

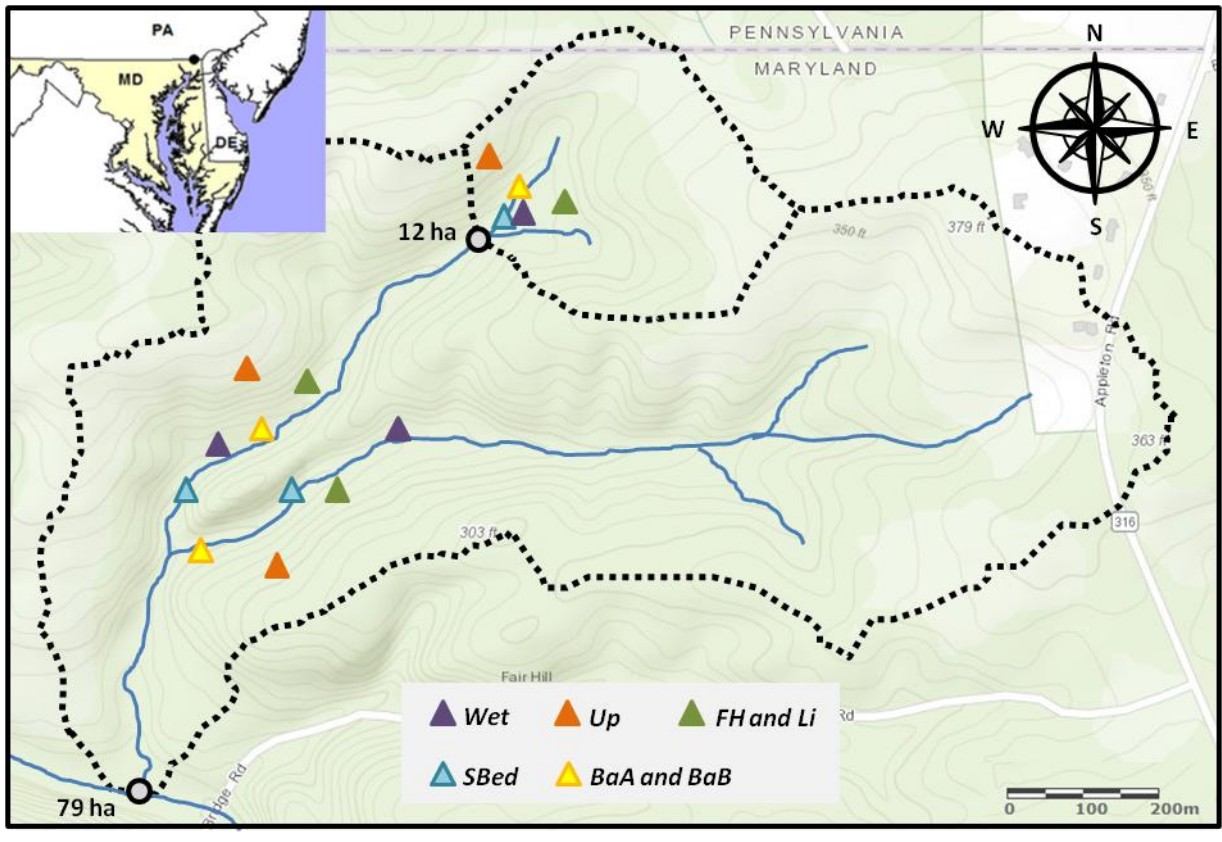







**Figure 02**

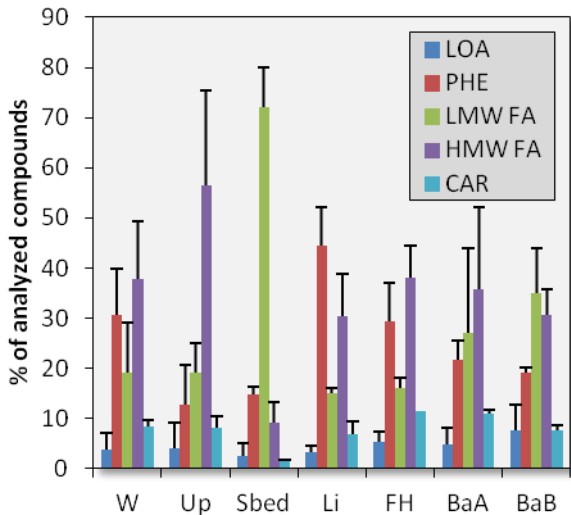





**Figure 03**

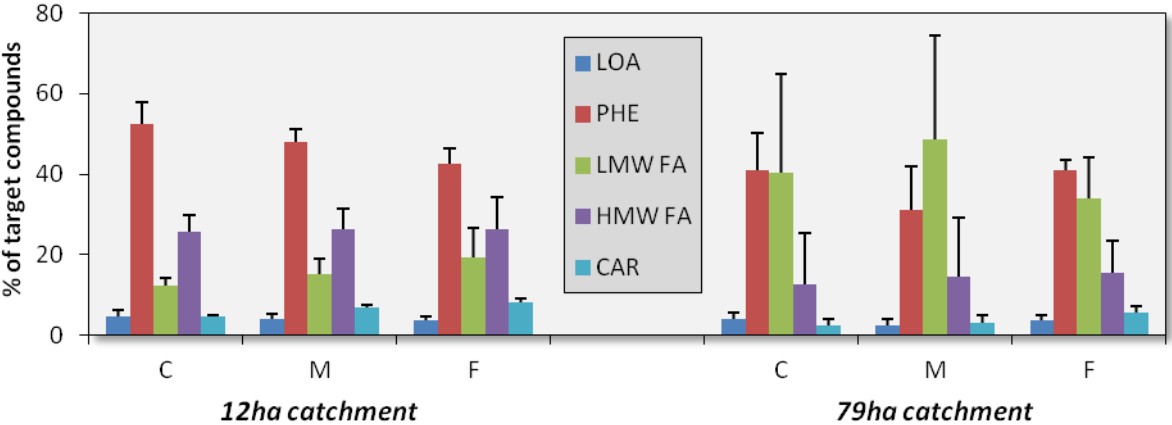





**Figure 04**

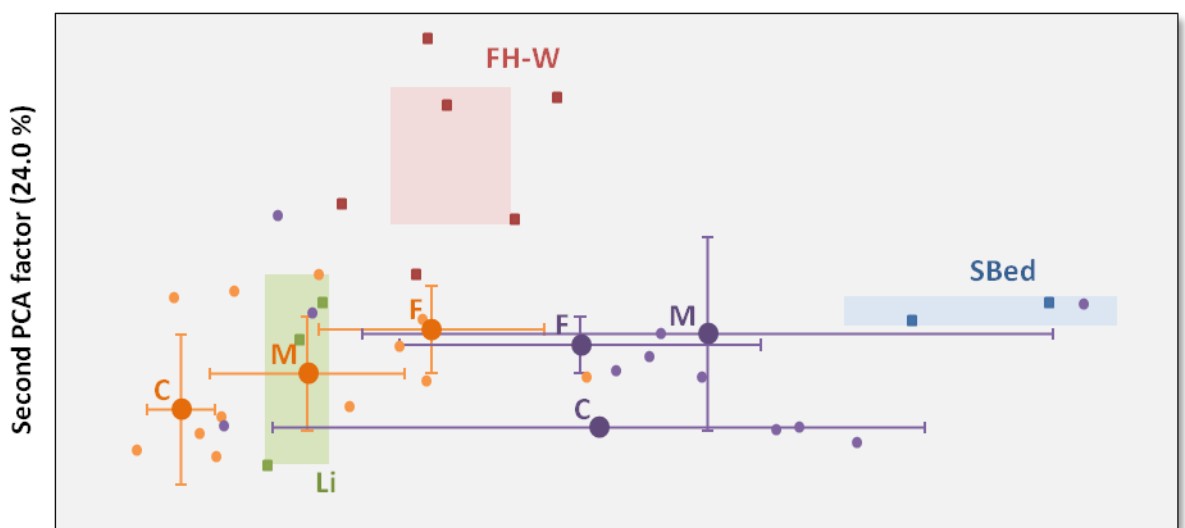



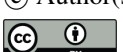



**Figure 05**

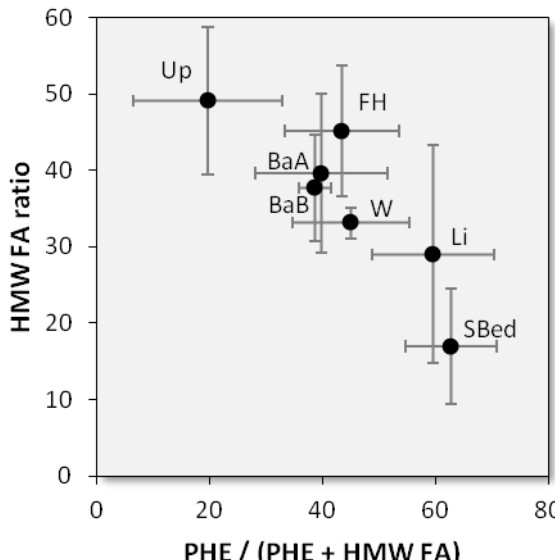





**Figure 06**

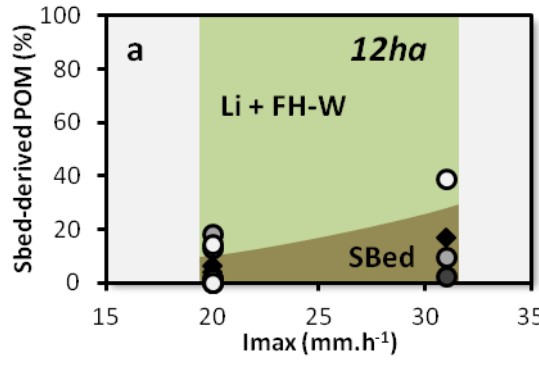

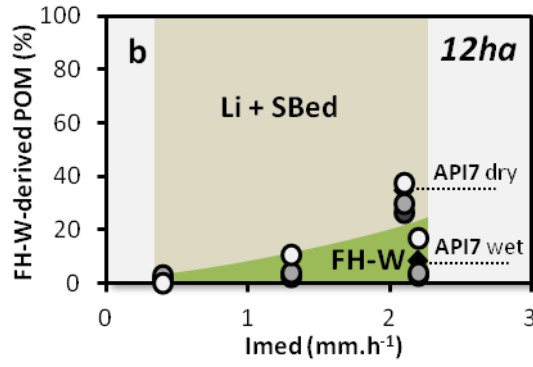

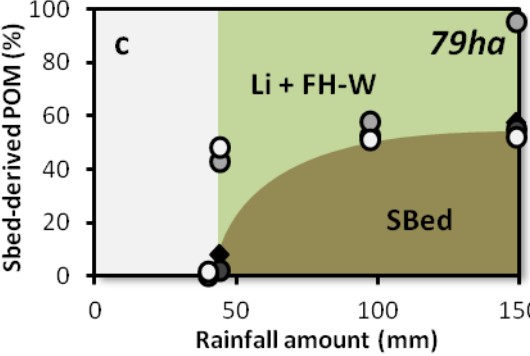


