# Peer review of "Molecular fingerprinting of particulate organic matter as a new tool for its source apportionment: changes along a headwater drainage in coarse, medium and fine particles as a function of rainfalls"

_Biogeosciences, 2017_

## Referee Comment (RC1) · Anonymous Referee #1 · 3 Oct 2017

The authors present an interesting study on the suitability of molecular analysis of POM using THM-GC-MS to investigate the sources of POM. The concept and implementation and analytics of the study is highly demanding and the approach very ambitious. While the results are interesting and successfully attribute the sources to litter, surface soils and in-channel sediments, the question arises why this technique is not used to differ between different litter types and soils. It is not so surprising that most of the POM derives from litter, more interesting would be to see from which land use types

in the catchment, which soils and vegetation surfaces. The manuscript is generally very well prepared, I no major comments. However, pretty difficult to read, because it is so full of abbreviations that you actually would need a permanent online translator to reat it. E.g.: "The relative proportion of LIG compared to HMW FA plotted against the proportion of ïＡｑïＡňïＡů-diacids and ïＡů-hydroxyacids with more than 20 C atoms among HMW FA resulted in a visual differentiation of Li and SBed from W, FH, BaA and BaB and from Up (Figure 5)." The manuscript is full of sentences like that. May be you could at least spell out the soils and horizons/layers and may be even different fractions. It would make the manuscript for sure easier to read. Not clear to me, what the 71 variables were, which were used in the PCA? Please make sure that all Figure captions and table titles are self-explaining, they are not at the moment. Partly again because of too many abbreviations which are not explained in the headings (e.g. Table 1 and 2: abbreviations not explained. Also Figure 2 BaA, BaB etc. . . . . . .).

---

## Referee Comment (RC2) · Anonymous Referee #2 · 8 Oct 2017

The authors display an interesting study on the use of molecular biomarkers to "apportion" the sources of particulate matter exported by storm events at catchment scale within two nested sub-catchments. The present study is complementary to a recently published paper by Rowland et al. (2017, cited by the authors), carried out on the same catchments but focussed on the bulk properties of particulate matter. Bulk organic matter approaches are not always successful as parameters such as total organic C - total N concentration and stable C and N isotope measurements, may provide equivocal

information, due for example to overlaps of end-members' signatures.

Molecular fingerprinting of organic matter, the approach reported by the authors in the present paper, is another possible way to strengthen source identifications, either as an independent tool or as a complementary discriminating approach. It is, however, a major challenge and any achievement in this direction is most welcome. Difficulties arise from many factors such as the heterogeneous composition of organic matter in the source end-members, the dynamics of particles' detachment, transport and mixing along catchment slopes, the characteristics of the storm event responsible for particulate matter export (i.e., temporal change in source supply during storm flows), landuse in the catchment or the distribution of storm flow events. Another difficulty comes from the analytical approach "in itself" that generates hundreds of molecular compounds. Accordingly, apportionment of source contributions is a very difficult goal as some of these compounds may occur in the composition of several source end-members. The accuracy of the source identification in the sediments may also be bias by the representativeness of the molecules in the samples. At last one additional level of difficulty is the use of a composite sediment sampling procedure that smoothes out differences in suspended matter composition during sediment transport.

The manuscript is well written and the authors fulfil several of these queries. The use of all peak areas provides a realistic picture of the composition of total organic matter in the samples. Statistical treatment of the data "sounds" accurate and the conclusions are consistent with usual interpretations in storm flow studies, i.e. enhanced deposition of litter at the upstream location, increasing proportion of in-channel material downstream or sediment sorting with increasing catchment's size (slope length). I do not have any major comments, only minor remarks that are reported below.

- Although the term is widely used in literature I think that the authors could briefly define or provide a reference for what they consider as particulate organic matter (POM). Does the term refer to "pure particulate organic matter" such as vegetation, root or leaf debris derived from litter or to one of the "soil and sediment matter properties", for example the concentration of organic compounds adsorbed or bound to mineral matter (clays, oxides,...) or both. Showing the parameters used to characterize bulk end-member and sediment compositions (in Rowland et al., 2017) may help. They could be added in Table 1 or reported in a new figure.

- I suggest a sharp reduction of the number of acronyms used in the text and a systematic report of the definition of the remaining ones in the legend of the figures. It would really help the reader.

---

## Author Comment (AC1) · 10 Oct 2017

Dear colleague, Many thanks for having taking the time to review our manuscript. I understand your idea of using this molecular fingerprinting method to differentiate between different litter types and soils. In this present paper, the sampling strategy was not design to investigate different litter types. However this method can be useful if there are important differences between plants in the catchment like gymnosperm versus angiosperm. Such situations could be interesting to test this tool. Differentiating between soils is in the scope of this paper since forest, wetland and upland soils are used as potential end-members. Since the vegetation is quite homogenous in the riparian areas of this catchment, forest and wetland have a similar molecular fingerprint while upland is clearly differentiated. About your difficulty to read the manuscript, we will make the appropriate modifications in order to improve its understanding in the text, figure captions and table titles. About your question on the 71 variables used in the PCA. With this method, we analyzed 112 target compounds and their relative proportions (using relative distribution is necessary to compare organic rich and organic poor samples) are used as variables for the principal component analysis (PCA). Since some targets are correlated or appeared in low abundance, the set of variable was reduced to keep only the most representative, without correlation between variables. These are the 71 variables that will be identified on table S1. Once again, thank you for your time in reviewing this manuscript. Sincerely

On behalf of the coauthors, Laurent Jeanneau

---

## Author Comment (AC2) · 16 Oct 2017

Dear colleague, Many thanks for having taking the time to review our manuscript. In this study, particulate organic matter was defined as the organic matter in the objects (natural debris, soil particles, colloids) that were trapped in the sampling system. The slots on the samplers were approximately 1.5 cm which represents the higher thresh-old. The samples collected there were dried before further analysis and then included

the smallest fractions defined as colloidal OM and dissolved OM. These precisions will be added in the manuscript. About your second suggestion on the sharp reduction of the number of acronyms, the manuscript will be changed in this way in order to help the reader. Once again, thank you for your time in reviewing this manuscript. Sincerely

On behalf of the coauthors, Laurent Jeanneau

———————————————

---

## Referee Comment (RC3) · Anonymous Referee #3 · 21 Nov 2017

Jeanneau et al. present an interesting study of the molecular composition of particulate organic matter in a small, low-relief, forested watershed. They first characterize potential endmembers, then use statistical analysis to apportion their contribution in suspended sediments collected under various runoff conditions. They compare their results with those obtained using elemental and isotopic composition (from a study recently published in Biogeochemistry) and discuss the impact of changing rainfall parameters on the composition of particulate organic matter exported by the river. Overall this is an interesting, well written study presenting novel data and proposing a new source apportionment approach. I do have numerous questions and comments, listed bellow, which I hope will help the authors tighten their manuscript during the revision process.

General comments:

1) Although the manuscript is generally very well written, the (very) extensive use of acronyms is really annoying.

2) Selection of the endmembers: I find strange to consider bed sediments as an endmember as they are in the first place derived from erosion processes and as such should be composed (at least in part) of a mixture of the other likely endmembers. And that is indeed supported by the data presented in the manuscript, e.g. L275-276.

It would also be good to describe the potential endmembers in greater details. What is currently provided section 2.2. is extremely succinct!

3) It's currently very hard for the reader to get a good feeling for how rainfall and hydrology varied across the study period (section 3.1). A good figure would be worth all these words.

4) endmember contributions: the statistical analysis leads the authors to group the surface horizon of forest soils (FH) and wetland soils (W) into a single endmember. I understand why they're doing so from a pure statistical standpoint but I think it would be good to discuss this in terms of surface processes. In other words, does it make practical sense to group these two endmembers together?

Figure 4 has me perplexed by the composition of the coarse fraction in the 12ha catchment. It clearly lays outside the boundary of the mixing defined by the endmembers. It suggests that the composition of the litter endmember is likely not adequate to capture the full compositional diversity in the suspended sediment samples.

5) comparison of molecular and elemental/isotopic data: it would be good to provide

more details regarding the d13C modeling exercise. For instance, what's the d13C value of each of the endmembers and how did the author choose these values? Looking at the supplemental figure it seems that the model does OK for measured d13C values higher than -29.2‰ or so, but not so well for more negative values (i.e. the modeled d13C values are very flat for measured d13C values < -29.2‰. This should be discussed.

Also, I wonder why the authors haven't tried to compare the molecular data model to the elemental and isotopic data model (instead of comparing molecular data to molecular data + elemental and isotopic data). I think it would be cleaner.

6) As it stands, section 4.3 isn't very convincing as the relationships between composition and rainfall parameters are very tenuous. I suggest dialing it back some such that it doesn't look as speculative as it currently does.

Specific comments:

L82: by design the sampler, although quite clever, isn't isokinetic and as such likely fractionates the suspended load with respect to grain size distribution. Perhaps add a note of caution in the text.

Section 3.1: the use of "mean" is a bit strange, e.g. in "mean median intensity". I know what the authors mean (no pun intended) but it sounds weird. Consider rewording.

L177-180: could this sample have unusually high relative proportions of petrogenic carbon? Is its stable isotope composition any unusual compared to the rest of the dataset?

L200-201: what's the source of Benzoic acid then?

L208-209: it seems to me that C16 and C18 could also be plant derived, at least in part. They're the most abundant FA in living plants.

L381: "mandatory" is weird in this context, consider rewording.

Data availability: There is no reason not to provide the data as a supplement. I urge the authors to do so.

---

## Author Comment (AC3) · 12 Dec 2017

Dear colleague, Many thanks for having taking the time to review our manuscript. In the following, we tried to answer your comments and to explain how we will consider them. Your first comment was on the extensive use of acronyms. The two other reviewers and you are unanimous on this point; some acronyms will be removed to improve the readability. Then you questioned the selection of end-members and specifically

our choice to consider bed sediments. Bed sediments were chosen as a potential end member because of the finality of the study. Among other, one goal is to determine the origin of POM exported during storm events. It would not imply the same conclusion in term of catchment management if the POM comes from bed sediments or from surface erosion. As a consequence it is necessary to consider it in the study. Moreover we generally agree that bed sediments could be a mixture of other end members. However, it needs to be recognized that end-member signatures could be further processed/modified while in the fluvial network since stream is not a passive pipe and thus acquire a unique signature? To account for this potential variability we considered the bed sediment as a potential separate source. Bed sediment could be a substantial store in the fluvial system, and to account for this large pool we have also characterized it separately. Âă About the description of the end-members, all details can be found in the publication by Rowland et al., 2017. Although we understand the interest for the reader to have this information in this article, since it has already been described, we prefer not detail it here to keep the size of the article reasonable. Âă Your third comment was on the description of the rainfall event. You suggest a figure could be a better option than the description (section 3.1). We choose in the paper to present the hydrologic – rainfall data with table 1 along with the description. Moreover a figure can be found in the paper published by Rowland et al., 2017. Âă About the end-member contributions, you ask if it makes practical sense to group forest floor organic horizon and wetland soil surface horizon. We think yes, there may be a practical sense because the vegetation is quite similar for those two areas, so the plant-derived contribution through roots will have similar composition. Moreover they have similar proportions of microbial chemical markers (13% of analyzed compounds in the wetland soil and 11% in the forest soil). Consequently their chemical compositions are close and they group in the statistical treatment. Âă Your next comment was on the figure 4 and the fact that some deposited sediments plot outside the triangle defined by end-members and you suggest that maybe litter end-member did not capture the full compositional diversity of the catchment. We agree with this point. The fact is that end-members molecular
and isotopic compositions were measured on the bulk sample. This is one limitation of the present approach, since during the erosive transfer of an end-member, there may be a size fractionation. This could explain why some samples plotted outside the area defined by end-members. To prevent from this limitation, end-members should be size fractionated and each fraction should be analyzed. Âă About the comparison between molecular and isotopic data, you asked more details about the modeling exercise. This exercise was simply an end-member mixing approach and we will precise at the end of section 2.4, line 154. Then you ask what are the values of d13C used for each end-members. They come from the sister study (Rowland et al., 2017), which is indicated line 154. About this comparison you mentioned a potential bias for more negative values. Looking at this figure, we can have this feeling. So I come back to the data and checked the residual from the linear regression model. The highest residual was for the extrem point (-29.76; -29.32 : measured; modeled) but the mean residual for d13C < -29.2 ‰ (n=6) was 0.287 ± 0.085 (mean ± SE) and for d13C > -29.2 ‰ (n=14), it was 0.220 ± 0.044. The deviation was not statistically different for lower d13C values. Âă On this comparison between molecular and isotopic/elemental data you ask why we did not try the elemental/isotopic data model alone. It was not performed because with four variables (d13C, d15N, TOC and N, [C/N being a linear combination of two variables]) it is not possible to differentiate between more than 5 sources using this type of statistical treatment (Walling, 2013). "As a minimum, n −1 properties are required to discriminate rigorously between n sources. Additional properties are frequently required to increase the reliability of the results." Âă Then you highlight the fact that the discussion in section 4.3 is based on fragile relationship because of only four events were investigated. We totally agree with this point that is the reason why two sentences have been inserted at lines 322 and 332 to precise that this part of the discussion is speculative. It is clear that future investigations are necessary to support this part of the discussion but we found it interesting enough to be mentioned. Âă Then you provide 7 specific comments: Âă 1.ÂăÂăÂăÂăÂăÂă About the design of the sampler. Âă Thank you for this comment. You are right, this method induces modification of

the velocity profile around the sampler, which could result in grain size fractionation. A sentence will be added to precise this point. Âă 2.ÂăÂăÂăÂăÂă The word "mean" will be replaced by "intermediate".

3.ÂăÂăÂăÂăÂă Could the stream bed sediment characterized by low amount of identified marker contain petrogenic OM? Âă To answer this point, we searched for petrogenic biomarkers such as n-alkanes, hopanes and steranes. Those compounds were not detected in this sample. Its isotopic fingerprint was -28.7 ‰ which is in the range of the values recorded for stream bed sediments in this catchment from -27.7 to -29.4 ‰ mean = -28.5 ‰ (n=5).Âă Âă 4.ÂăÂăÂăÂăÂă What is the source of benzoic acid? Âă The proportion of benzoic acid in soil profiles increased with depth which has been interpreted as a consequence of the humification process (Chefetz et al, 2000). However, the humification of organic matter as a biogeochemical process has been clearly questionned this last tenth of years. We can assume that, since its evolution is inversely correlated to lignin phenols (slope = -0.32; $r^2$ = 0.23; p-value = 0.001) it derive from the degradation of tannins and lignins. Âă 5.ÂăÂăÂăÂăÂă C16:0 and C18:0 alkanoic acids may be consider as plant-derived. Âă Yes, C16:0 and C18:0 are ubiquitous and may derive from microbial and plant-derive inputs. For this reason they are not used for the calculation of microbial markers (Jeanneau et al., 2014). A precision will be added in the text to mention it. Âă 6.ÂăÂăÂăÂăÂă The word "mandatory" will be replaced by "necessary". Âă 7.ÂăÂăÂăÂăÂă We agree that data should be freely accessible and we will prepare the data to add them in supplementary materials. Âă Once again many thanks for your time and consideration in reviewing our paper. Sincerely Âă Laurent Jeanneau On behalf of the coauthors Âă Âă Chefetz, B., Chen, Y., Clapp, C. E. and Hatcher, P. G.: Characterization of Organic Matter in Soils by Thermochemolysis Using Tetramethylammonium Hydroxide (TMAH), Soil Sci Soc Am J, 64(2), 583–589, 2000.

Jeanneau, L., Jaffrezic, A., Pierson-Wickmann, A.-C., Gruau, G., Lambert, T. and Petitjean, P.: Constraints on the Sources and Production Mechanisms of Dissolved Organic Matter in Soils from Molecular Biomarkers, Vadose Zone J., 13(7),

doi:10.2136/vzj2014.02.0015, 2014.

Rowland, R., Inamdar, S. and Parr, T.: Evolution of particulate organic matter (POM) along a headwater drainage: role of sources, particle size class, and storm magnitude, Biogeochemistry, 133(2), 181–200, doi:10.1007/s10533-017-0325-x, 2017.

Walling, D. E.: The evolution of sediment source fingerprinting investigations in fluvial systems, J. Soils Sediments, 13(10), 1658–1675, doi:10.1007/s11368-013-0767-2, 2013. Âă Âă Âă Âă Âă Âă

---

## Author Response (AR1)

Dear Dr. Battin,

I wish you an happy new year and all the best for 2018.

You will find the new version of the manuscript « Molecular fingerprinting of particulate organic matter as a new
tool for its source apportionment: changes along a headwater drainage in coarse, medium and fine particles as a
function of rainfalls » modified according to the comments of the three anonymous reviewers.

In the following you will find a descrption of the modifications that were made. They are highlighted in  green in
the text. I think that they have improved the quality and the readability of my paper.

Have a good day

Sincerely

Laurent Jeanneau

*On behalf of the coautors*

Throughout the text : the acronyms for fatty acids (FA) and phenolic compounds (PHE) were removed. From the
beginning of the discussion section, the signification of the acronyms of the end members was reminded.
(Reviewers 1, 2 and 3)

In Table S1, the variables used for the statistical treatment were identified with the symbol *. (Reviewer 1)

The caption of the figures and tables have been modified to be self-explaining. (Reviewer 1)

The definition of what is POM in this study was added at paragraph 2.2. (Reviewer 2)

Two assumptions regarding the fact that some samples plotted outside the end-members triangle on figure 4 were
added at the end of the first paragraph of the section 3.5. (Reviewer 3)

About the comparison between molecular and isotopic data, a precision about this exercice was added at the end
of section 2.4. « using an end-member mixing approach » (Reviewer 3)

The following sentence :Such a method induces modification of the velocity profile around the sampler, which
could result in grain size fractionation. was added in section 2.2 to precise that the sampler is not isokinetic
(Reviewer 3; specific comment 1)

The word "mean" was replaced by "intermediate" in the section 3.1 for the description of Event 1 E1. (Reviewer
3; specific comment 2)

A precision about $n$-$C_{16:0}$ and $n$-$C_{18:0}$ that can derive from plant-derived inputs was added at the end of the section
3.3 (Reviewer 3; specific comment 5)

Mandatory was replaced by necessary at the end of the first paragraph of section 4.4. (Reviewer 3; specific
comment 6)

The data are now available in the supplementary files. (Reviewer 3; specific comment 7)

[revised manuscript text omitted]